

# The impact of bedrock meander cutoffs on 50 ka-year-scale incision rates, San Juan River, Utah

Aaron T. Steelquist[1], Gustav B. Seixas[2], Mary L. Gillam[3], Sourav Saha[4], Seulgi Moon[5], and George E. Hilley[1]

[1]Department of Geological Sciences, Stanford University, 450 Jane Stanford Way, Building 320, Stanford, California, U.S.A., 94305
[2]Skagit River System Cooperative, La Conner, Washington, U.S.A., 98257
[3]115 Meadow Road East, Durango, Colorado, U.S.A., 81301
[4]Kentucky Geological Survey, University of Kentucky, Lexington, Kentucky, U.S.A., 40506
[5]Department of Earth, Planetary, and Space Sciences, University of California Los Angeles, Los Angeles, California, U.S.A., 90095

**Correspondence:** Aaron T. Steelquist (steelqat@stanford.edu)

**Abstract.** Incision rates of major rivers may reflect the effects of drainage reorganization, hillslope processes, tectonic uplift, climate, the properties of rocks into which rivers incise, and other autogenic processes. On the Colorado Plateau, incision rates along the Colorado River have been interpreted as resulting from abrupt base-level changes produced by the integration of the Colorado River system. Specifically, the integration of the Colorado River in the location of Grand Canyon is thought to have created a knickpoint, enhanced by lithologic contrasts, which is retreating upstream. While evidence exists for a <1 Ma acceleration of incision on parts of the Colorado River, uncertainty about the processes reflected in shorter-term incision rates muddies comparison with longer-term averages. In this work, we combine a cosmogenic radionuclide depth profile exposure age and post-Infrared Infrared Stimulated Luminescence (p-IR IRSL) to date fluvial deposits adjacent to the San Juan River, a major tributary of the Colorado River, near Mexican Hat, Utah. The deposits, resting on a 32 m strath surface, are constrained to be ∼28–40 ka, suggesting an incision rate of 804–1151 m Myr⁻¹, nearly an order of magnitude higher than the long-term rate of ∼140 m Myr⁻¹ over the past ∼1.2 Ma. We observe fluvial deposits that were abandoned due to a bedrock meander cutoff, which partially explains our accelerated incision rate. We use a simple geometric model, informed by our field data, to demonstrate how planform river evolution may, in some circumstances, increase short-term incision rates, relative to long-term incision rates. These short-term rates may also reflect a combination of autocyclic and climatic processes, which limits their ability to resolve longer-term changes in incision rate that may be related to changes in base-level or tectonics.

## 1 Introduction

The landscape of the Colorado Plateau in the American Southwest is known for deep canyons, broad plateaus separated by steep escarpments, and mesas that protrude from otherwise low-relief surfaces. Regional denudation and the resulting landforms are closely tied to the rate of incision by large rivers, which serve as major conduits for mobilized sediment. Despite the centrality of the Colorado River and its tributaries to understanding the complex geologic, tectonic, and climatic history that shaped the



Colorado Plateau, significant uncertainty still exists around the fluvial evolution of the region and how the major rivers came to their present configuration.

Integration of the Colorado River and its tributaries is generally believed to have occurred five to six million years ago when a combination of integration through Grand Canyon (Karlstrom et al., 2014), basin spill-over (Blackwelder, 1934; Pearthree and House, 2014), and groundwater sapping (Crossey et al., 2015) connected the Colorado Plateau to the Gulf of California. Reconstruction of how tributaries became integrated and responded to downstream changes in base level relies on analysis of dateable geologic material. These alluvial and fluvial deposits have been used to constrain the incisional history and evolution of the Green (Aslan et al., 2010; Darling et al., 2012), Upper Colorado (Aslan et al., 2019), and Little Colorado Rivers (Karlstrom et al., 2017) over the last 1.5 Ma. They have been used to infer the regional tectonic evolution of the Colorado Plateau and the Rocky Mountains (Karlstrom et al., 2012; Pederson et al., 2013; Rosenberg et al., 2014; Aslan et al., 2019). For example, Darling et al. (2012) synthesized multiple decades of studies documenting incision on the Colorado River and its tributaries. Over the >1 Ma timescale, incision rates below a knickpoint at Lees Ferry, Arizona are ~170–230 m Myr[-1] (Pederson et al., 2002; Karlstrom et al., 2008; Polyak et al., 2008), higher than the ~110–130 m Myr[-1] rate above the knickpoint (Darling et al., 2012). These spatial patterns have been interpreted to represent a change in base-level conditions following integration through Grand Canyon (Wolkowinsky and Granger, 2004; Cook et al., 2009), regional tectonism (Karlstrom et al., 2008, 2012), or contrasts in rock properties (Bursztyn et al., 2015).

Over shorter timescales (<0.5 Ma), incision rates at Lees Ferry are estimated to be ~350 m Myr[-1] over 150 ka. This higher incision rate, compared to the Ma-scale rate, has been attributed to isostatic feedback from drainage integration (Pederson et al., 2013), although this is contested by Karlstrom et al. (2013). Short-term incision rates along the Upper Colorado River have also yielded higher rates than those averaged over the >1 Ma timescale, with some as high as >3700 m Myr[-1] where drainage reorganization appears to play a role (Aslan et al., 2019). Others have noted that short-term incision rates can be particularly sensitive to glacial oscillations, which contribute to fluctuating sediment supply, discharge, and isostatic effects in rivers with montane or alpine headwaters (Hancock and Anderson, 2002) as well as the cyclical nature of bedrock incision and aggradation in rivers (Finnegan et al., 2014).

This work focuses on the incision history of the San Juan River, which is a major tributary in the Colorado River system. Lucchitta and Holm (2020) proposed this tributary reflects an upstream expansion of the Colorado River system following its integration. In this view, an early course of the lower San Juan River formed by exploiting a posited Oligocene to late Pleistocene "Crooked Ridge River" that flowed through the Kletha Valley (Cooley et al., 1969; Hunt, 1969), although this has been rebutted by Heizler et al. (2021) based on ages of deposits that are inconsistent with that location for a paleo-San Juan River.

Discerning the recent incision history of the San Juan River is challenging due to a dearth of dated fluvial or alluvial material in the region. Two notable exceptions to this are the cosmogenic [26]Al–[10]Be burial dating of sediments near Bluff, Utah (Wolkowinsky and Granger, 2004), which were also recently dated using detrital sanidine (Heizler et al., 2021) and the exposure age dating of surfaces grading to the present location of the San Juan River along the flanks of Navajo Mountain (Garvin et al., 2005). These incision rate estimates at Bluff (110 ± 14 m Myr[-1] and 140 m Myr[-1] for cosmogenic isotopes





and detrital sanidine, respectively) and at Navajo Mountain (∼400–825 m Myr⁻¹) are separated by ∼120 km, employ different surface exposure dating methodologies, and sample different time intervals of the incision history of the San Juan River. This has led to some uncertainty as to whether the differences in incision rates between the two sites represent a methodological bias, a regional spatial variation in incision rate, a regional temporal fluctuation in erosion rate (Darling et al., 2012), or a bias toward higher incision rates over shorter timescales (Finnegan et al., 2014).

Interestingly, Pleistocene river terrace surfaces suitable for constraining the short-term incision rate of the San Juan River downstream of the Bluff site exist, but few have been characterized or dated in the scientific literature. This presents an opportunity to constrain the incision history of the San Juan River over different periods of time at two proximal sites, and to understand the mechanisms that may produce this incision history. This study uses geomorphic mapping, in-situ cosmogenic ¹⁰Be depth profile analysis, and post-Infrared Infrared Stimulated Luminescence (p-IR IRSL) dating methods to constrain the rates and mechanisms that produce the short-term incision history along the San Juan River near Mexican Hat, Utah, ∼45 km downstream from Bluff. We find that deposition of the terrace materials occurred during or after 39.8 ± 3.1 ka, which is consistent with the ¹⁰Be depth profile exposure age of 27.9 ± 0.8 ka. The implied ten-thousand-year incision rate measured in this study (∼804–1151 m Myr⁻¹) greatly exceeds those measured over ∼1.20-1.63 Ma timescale (84-140 m Myr⁻¹). The modern San Juan River bypasses the dated material, which lie in a cutoff meander entrenched into resistant bedrock. We establish a conceptual model for how meander cutoff can result in a discrepancy in the apparent long-term and short-term incision rates and explore the effect of lithology on formation of planform erosional features which can preserve datable geologic material, with implications for biases when measuring vertical incision.

## 2 GEOLOGIC SETTING

### 2.1 Regional geology

The San Juan River is a major tributary of the Colorado River and drains ∼64,000 km² in southeast Utah, northeast Arizona, northwest New Mexico, and southwestern Colorado (Figure 1). The headwaters of the San Juan River are in the formerly glaciated San Juan Mountains, from which the river flows ∼600 km to its confluence with the Colorado River at Glen Canyon. The upper San Juan River occupies a broad valley attributed to Miocene incision (Cooley et al., 1969; Hunt, 1969) though recent controversy about the evolution of the San Juan River has brought into question how and when the San Juan River established its modern course (Hereford et al., 2016; Lucchitta and Holm, 2020; Heizler et al., 2021).

This broad valley becomes inset by a narrow gorge just downstream of Chinle Creek, Utah that extends to the confluence with the Colorado River at Glen Canyon, Arizona (Figure 1). This gorge runs through the Monument Upwarp, a large structural fold comprised dominantly of Pennsylvanian and Permian marine sedimentary rocks with discrete zones of Laramide-style, fault-cored monoclines and anticlines with corresponding synclines (Ziony, 1966). The San Juan River incises through a number of these features, including Comb Ridge monocline, which bounds the Monument Upwarp on the east, and Raplee Ridge monocline (Figure 1). Downstream of Raplee Ridge, the deeply incised meanders of the Goosenecks led early researchers to infer that the San Juan River predated the Laramide Monument Upwarp (Wengerd, 1950). However, recent work has shown



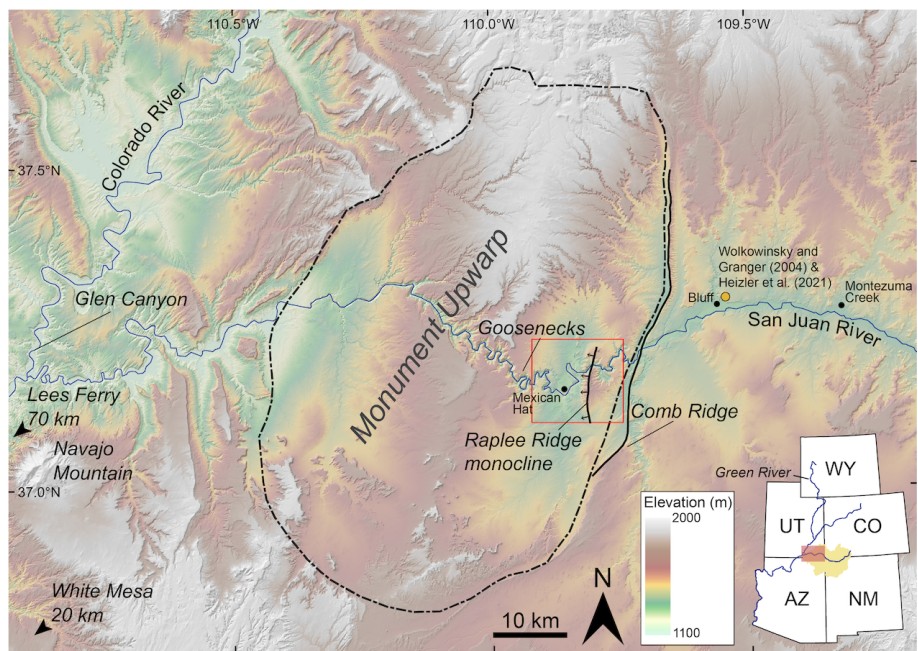

**Figure 1.** Shaded-relief map of the Four Corners region (southwest Utah). Major physiographic, geographic, and fluvial features are labeled. Inset shows San Juan drainage in yellow polygon with Colorado River and Green River shown in blue lines. Hollow red box in detailed map shows area of Figure 2. AZ – Arizona, CO – Colorado, NM – New Mexico, UT – Utah, WY – Wyoming.

that incising river meanders are not exclusively inherited from preceding alluvial rivers but may form and change shape in
actively incising bedrock settings (Hovius and Stark, 2001; Finnegan and Dietrich, 2011; Limaye and Lamb, 2014).

We focus on the San Juan Canyon between Raplee Ridge monocline and Gooseneck State Park, near Mexican Hat, Utah (Figure 2). Raplee Ridge is a ∼14 km long monocline with a north-south-oriented, doubly plunging fold axis. The fold forming Raplee Ridge has been variably described in the literature as a monocline and anticline, though here we opt for Raplee Ridge monocline as the topographic expression and fault kinematics (Hilley et al., 2010) are similar to other Laramide monoclines in
the area. The core of the monocline is comprised of interbedded erodible shale and resistant limestone layers of the Pennsylvanian Rico and Paradox Formations (Figure 2). Downstream of Raplee Ridge, the San Juan River is bounded by the Permian Halgaito Tongue Formation mudstone for ∼11 km before again incising into the Rico and Paradox Formations as it enters the Goosenecks west of Mexican Hat, Utah.

## 2.2 Sample site - Mexican Hat quarry

We focused our study on the area surrounding a gravel quarry 1.6 km northeast of Mexican Hat, Utah, north of the San Juan River (Figure 2 and Figure 3a). This quarry was active at the time of sampling but is now inactive. O'Sullivan (1965) briefly described these fluvial deposits, saying "... there are at least four levels of terrace gravels, but all cover small areas.... Parts of



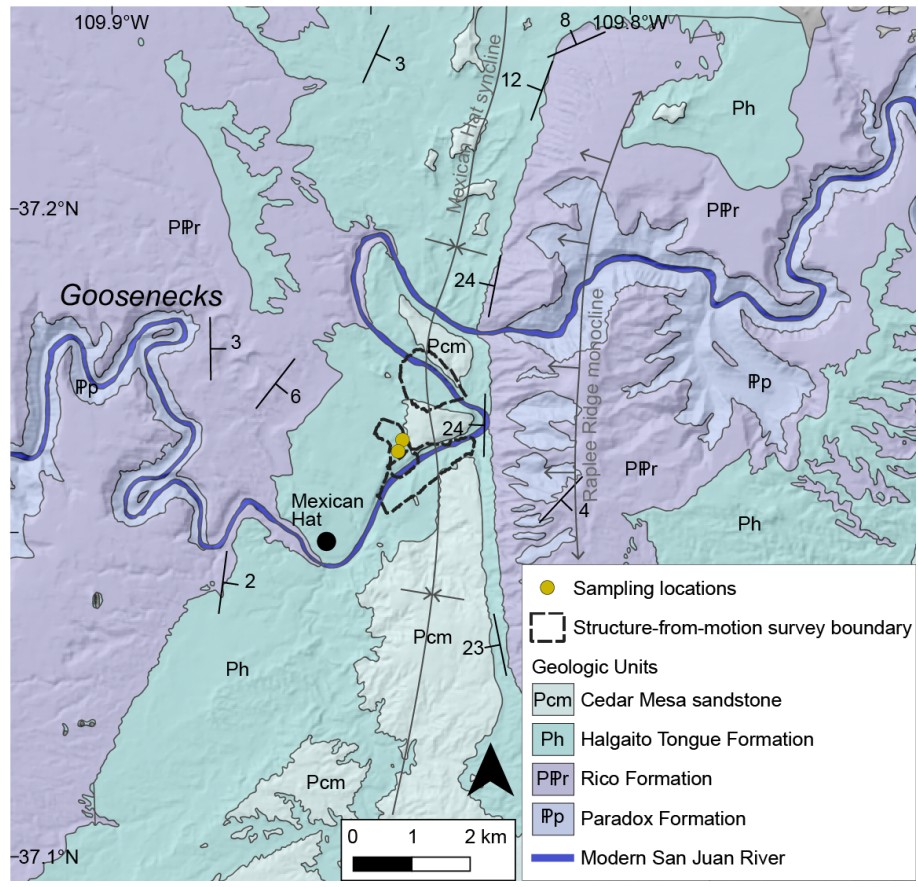

**Figure 2.** Geologic map of the Mexican Hat, Utah region, modified from O'Sullivan (1965). Boundaries of structure-from-motion surveys and sampling locations in the center of the map.

the town of Mexican Hat are built on the most extensive terrace in this area, which is about 125 feet above river level." Figure 3d shows an example of the context of these deposits in the San Juan Canyon.

The fluvial deposits at the Mexican Hat quarry are part of a fill terrace resting on a strath surface 32 meters above the modern San Juan River surface. The lower part of the deposit is composed primarily of medium to coarse, clast-supported cobbles in a sand matrix (Figure 3b). Gravel deposits are increasingly undisturbed by mining activity towards the San Juan River, where the exhumed top forms a cobble-dominated gravel pavement. Gravel deposits are ∼3 m thick but thin towards a small gully that bisects the deposit.

The gravel quarry is bounded on the northeast by a mesa of Halgaito Tongue Formation and Cedar Mesa sandstone member of the Cutler Formation. At the base of this mesa are eroded remnants of sand deposits ∼4 m thick (Figure 3c). Sand deposits throughout the region are largely attributed to aeolian processes (O'Sullivan, 1965), however this sand variably overlies the sandy gravel and bedrock strath surface. Exposures contain subhorizontal beds with plane bedding, relatively massive upward





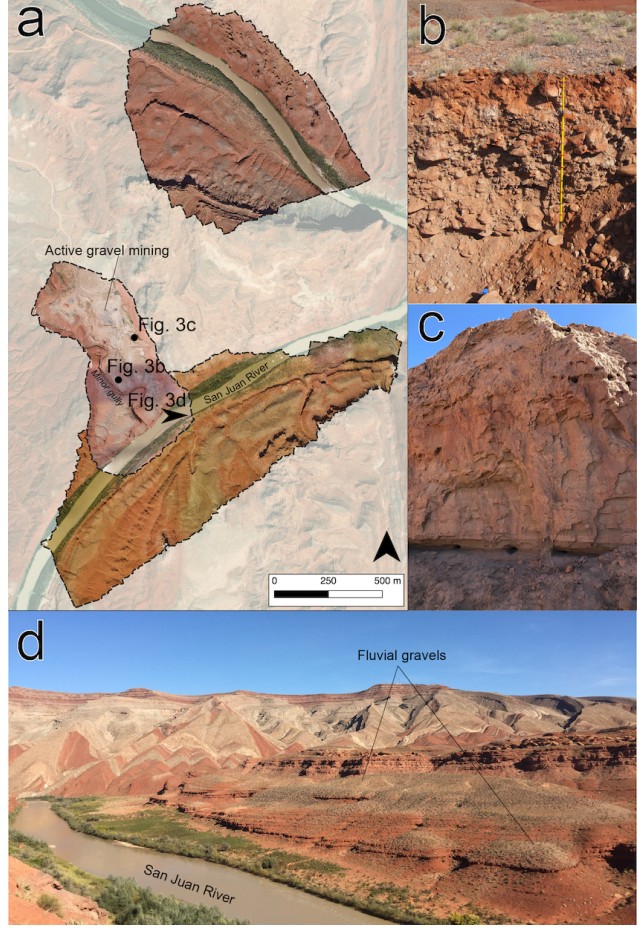

**Figure 3.** Figure 3: a) Orthographic images created using structure-from-motion techniques, including approximate image locations (black dots) and look directions (black arrow) of subsequent panels b) Fluvial gravels sampled for [10]Be analysis. Outcrop height in image is approximately 2 m. c) Poorly consolidated fluvial sand outcrop sampled for IRSL dating, resting on fluvial gravels; sand height in image is approximately 3 m. d) View looking approximately east from the gravel quarry across the San Juan River. Note multiple bedrock straths (red-brown) mantled by fluvial gravels (gray). Fluvial sands overlying the higher gravels appear as light tan patches where not hidden by colluvium. Raplee Ridge is in the background.

fining units, ripples, and silt drapes, which suggest a fluvial origin. This is further bolstered by the absence of high-angle cross
bedding that would indicate aeolian deposition. The sand appears inset into the gravel at some locations, plausibly filling a paleochannel in the gravel top; however, disruption by quarrying operations mean this relationship cannot be traced throughout the area. Similar sandy deposits are more widespread nearby, suggesting sand deposits originally covered the sandy gravel at the quarry, making the gravel pavement an eroded rather than depositional surface. As cosmogenic radionuclide depth profile dating is sensitive to erosion of the deposit surface, which is possible given our observations, we used multiple dating
techniques to determine the deposit age.



# 3 METHODS

## 3.1 Structure-from-motion (SfM) surveying and geomorphic mapping

We completed three aerial surveys of the San Juan River directly downstream of Raplee Ridge using an Unmanned Aerial Vehicle (UAV). Each survey was processed in Agisoft Metashape (Version 1.6.5) to form three-dimensional Red-Green-Blue-labeled point clouds. To ensure consistency across surveys and proper georeferencing, we established ground control points using a high-precision GPS (lateral precision ∼5 cm, vertical precision ∼20 cm) for each survey region and used these points to reference image locations during Metashape processing.

We used the SfM point clouds to create paired Digital Elevation Model (DEM) and orthographic photos for each survey region. Using these data in combination with field observations, we mapped fluvial deposits and fill terraces within the San Juan Canyon between Raplee Ridge and Mexican Hat. All fluvial deposits include sandy pebble to cobble gravels, often with scattered boulders, that rest directly on bedrock strath surfaces. Some of these gravels are overlain by several meters of fluvial sands and slightly gravelly sands, resembling those at the quarry. We separated the relative age of fluvial deposits based on the height of the underlying strath surfaces above the modern San Juan River in the SfM datasets. We also mapped gravel deposits in the San Juan Canyon between Raplee Ridge and Mexican Hat using satellite imagery outside of the SfM survey regions. However, the low resolution of topographic data for the region (∼10 m per pixel and >1 m vertical accuracy) and human disturbance at Mexican Hat allow only speculative correlations to be made between these terrace levels and those capture by SfM.

## 3.2 Cosmogenic radionuclide dating - Reassessing Bluff terrace data

$^{10}$Be and $^{26}$Al are primarily produced through spallation of Si and O when material is exposed to cosmic-ray bombardment at (or near) Earth's surface (Lal and Peters, 1967; Nishiizumi et al., 1991), though some contribution from negative-muon capture and fast-muon reactions also occurs (Heisinger et al., 2002a, b). As neutrons rapidly attenuate with depth, the concentration of $^{26}$Al–$^{10}$Be in rock material can be used to determine the amount of time that material has been at the surface (Lal, 1991). For fluvially transported material, this is typically assumed to be the combination of $^{26}$Al–$^{10}$Be produced during erosion from its source region (typically referred to as the "inherited concentration") and $^{26}$Al–$^{10}$Be produced after terrace abandonment (Granger and Muzikar, 2001). However, if buried deeper than this attenuation depth, the relative abundance of $^{26}$Al in previously dosed sediment will decrease relative to $^{10}$Be because of the shorter half-life of the former relative to the latter (Lal, 1991). While individual clasts have different erosional histories, and thus different inherited $^{26}$Al–$^{10}$Be concentrations, well-mixed fluvial sands represent catchment-average denudation rates (Granger et al., 1996). Given this averaging effect, in assessing in-situ $^{26}$Al and $^{10}$Be, we assume sand sampled from multiple depths shares a single inherited concentration, simplifying the process of dating fluvial material.

Previous work at the Bluff site (Wolkowinsky and Granger, 2004) exploited both the near-surface accumulation of $^{10}$Be and $^{26}$Al, as well as the more rapid decay of $^{26}$Al at depth to jointly solve for the exposure age, erosion, and inheritance of clasts at the Bluff site. However, since the time of that initial study, the production and decay rates for these cosmogenic isotopes have



been revised (e.g., Lifton et al., 2014), thus we first updated the results using modern parameter values to ensure consistency
between these long-term, and our short-term incision rate estimates within this region of the San Juan River.

The procedure for computing ages and incision rates is documented in Wolkowinsky and Granger (2004). Briefly, we calculated expected concentrations of $^{10}$Be and $^{26}$Al, given a terrace age, terrace erosion rate, individual clast inheritance (assuming $^{26}$Al–$^{10}$Be production ratio of 7.112), bulk density, and loss of material from the terrace surface as follows (Wolkowinsky and Granger, 2004, their Equation DR1):

$$N = N_{inh}\, e^{\frac{-t}{\tau}} + \left[\frac{P_n e^{-\frac{\rho z}{\Lambda}}}{\left(\frac{1}{\tau}+\frac{\rho\varepsilon}{\Lambda}\right)}\right]\left[1-e^{-t\left(\frac{1}{\tau}+\frac{\rho\varepsilon}{\Lambda}\right)}\right] + \left[\frac{Y A_1 e^{-\frac{\rho z}{L_1}}}{\left(\frac{1}{\tau}+\frac{\rho\varepsilon}{L_1}\right)}\right]\left[1-e^{-t\left(\frac{1}{\tau}+\frac{\rho\varepsilon}{L_1}\right)}\right] +$$
$$\left[\frac{Y A_2 e^{-\frac{\rho z}{L_2}}}{\left(\frac{1}{\tau}+\frac{\rho\varepsilon}{L_2}\right)}\right]\left[1-e^{-t\left(\frac{1}{\tau}+\frac{\rho\varepsilon}{L_2}\right)}\right] + \left[\frac{B e^{-\frac{\rho z}{L_3}}}{\left(\frac{1}{\tau}+\frac{\rho\varepsilon}{L_3}\right)}\right]\left[1-e^{-t\left(\frac{1}{\tau}+\frac{\rho\varepsilon}{L_3}\right)}\right] \tag{1}$$

where $N$ (atm g$^{-1}$) is the measured concentration of $^{26}$Al or $^{10}$Be, $N_{inh}$ is the inherited concentration for $^{26}$Al or $^{10}$Be, $t$ (yrs) is the exposure age, $z$ (cm) is sample depth (allowing for addition material to all samples), $\tau$ (yr$^{-1}$) is the radioactive mean life of $^{26}$Al or $^{10}$Be, $P_n$ (atm g$^{-1}$ yr$^{-1}$) are the production rates of $^{26}$Al or $^{10}$Be by nucleon spallation, $\rho$ (g cm$^{-3}$) is the density of the
165 overburden, and $\varepsilon$ (cm/yr) is the terrace erosion rate. Following Wolkowinsky and Granger (2004), we used $\Lambda = 160$ g cm$^{-2}$, $A_1 = 170.6$ and $A_2 = 36.75$ at sea-level and high latitude, $L_1 = 738.6$ g cm$^{-2}$, $L_2 = 2688$ g cm$^{-2}$ and $L_3 = 4360$ g cm$^{-2}$, $Y_{Al} = 4.24$ x $10^{-3}$ and $B_{Al} = 0.192$ for $^{26}$Al and $Y_{Be} = 4.91$ x $10^{-4}$ and $B_{Be} = 0.023$ for $^{10}$Be. We updated production rates for the sample location and elevation from the time-dependent, nuclide-dependent scaling scheme of Lifton et al. (2014) estimated using the CRONUScalc online interface (Marrero et al., 2016). $A_1$ and $A_2$ were scaled for latitude and elevation using a factor
of 1.5231 (derived from CRONUScalc). We also adjusted the measured $^{10}$Be concentrations by a factor 1.0351 given the effect of the revised $^{10}$Be half-life from (Nishiizumi et al., 2007) on the standards against which the Bluff $^{10}$Be/$^9$Be were compared.

We then used the reported measured abundances of $^{10}$Be and $^{26}$Al to define a misfit between calculated $^{10}$Be and $^{26}$Al concentrations as follows:

$$\phi = \frac{1}{v}\left[\sum\left[\frac{N_{Al}^{pred}-N_{Al}^{meas}}{\sigma_{Al}^{meas}}\right]^2 + \sum\left[\frac{N_{Be}^{pred}-N_{Be}^{meas}}{\sigma_{Be}^{meas}}\right]^2\right] \tag{2}$$

where $\phi$ is the misfit, $v$ is the degrees of freedom (6, for these data), and $\sigma$ is the concentration measurement error for each element. We numerically minimized this misfit using a steepest-descent method to solve for the model parameters that best fit the observed data. As with Wolkowinsky and Granger (2004), we then set all model parameters equal to their best-fitting values, while varying terrace erosion rate and surface exposure age to provide an estimate of the uncertainties in these parameters. The uncertainties are equal to the minimum and maximum values of erosion rate and exposure age for which $\phi$ is within one
standard error of the best-fitting values.

### 3.3 Cosmogenic radionuclide dating - $^{10}$Be depth profile

We used in situ-produced $^{10}$Be in quartz grains to determine the age of the fluvial deposits at the Mexican Hat gravel pit. We sampled quartz sand at five depths ranging from 5 cm to 2 m below the present top of the gravel (each sample was



taken from a ∼2 cm horizon) and processed the samples at Stanford University using a modified procedure from Kohl and
Nishiizumi (1992). Full sample preparation description can be found in Supplemental Information. All samples were analyzed
using the accelerated mass spectrometer (AMS) at Lawrence Livermore National Laboratory in Livermore, California using
07KNSTD3110 standard with assumed $^{10}Be/^9Be = 2.85 \times 10^{-12}$.

We used the resulting $^{10}Be/^9Be$ ratio from the AMS analyzes to determine the concentration of $^{10}Be$ at each sample depth
and fit a line with the form of Equation 1 using nonlinear least squares minimization, solving for exposure age ($t$) and $^{10}Be$
inheritance ($N_{inh}$). We assumed no terrace erosion and used the bulk density calculated from our reassessment of the Bluff
terrace data.

### 3.4 Infrared stimulated luminescence (IRSL) dating

Minerals such as quartz and feldspar contain luminescence signals that result from radioactive decay in naturally occurring
radioisotopes in the surrounding material, with signal accumulating during burial (Huntley et al., 1985; Rhodes, 2011). As
sediment grains are transported by geomorphic processes, exposure to sunlight bleaches the luminescence signal. Thus, we can
use luminescence dating of buried sediments to infer the time since the last exposure to sunlight (Rhodes, 2011; Buylaert et al.,
2012). While previously applied to amalgamated fluvial sediment, recent advances in measuring single grain luminescence
signals in many individual grains taken from a single deposit yields a distribution of luminescence ages (Arnold et al., 2007;
Rhodes, 2015; Gray et al., 2018). The single grain age distributions give not only the last depositional age but potentially
provide information about the depositional setting (e.g., Saha et al., 2021). For example, aeolian transport will likely expose
all grains to sufficient sunlight to bleach all luminescence signals, resulting in a unimodal population of single-grain ages. In
contrast, transport in a silty fluvial setting like that in the San Juan River may leave a population of grains unbleached, leading
to a multi-modal age distribution due to residual luminescence from a more complex bleaching history (e.g., Gliganic et al.,
2017). Using single grain IRSL dating techniques allows confirmation of the depositional character of the sands at the quarry.

We used post-Infrared Infrared Stimulated Luminescence (p-IR IRSL, Reimann et al., 2012; Rhodes, 2015) analysis in
single-grain K-feldspars to determine the distribution of last sunlight exposure in grains within the poorly consolidated, fluvial
sand at the Mexican Hat quarry. We sampled sand ∼1 m above the gravel deposit by inserting an opaque 5-cm diameter metal
tube into a freshly cleaned face of the sand deposit and immediately capping it. K-feldspar grains of 175–200 $\mu$m diameter
from the sedimentary samples were isolated at the UCLA Luminescence Laboratory using the procedures in Rhodes (2015)
under dim amber LED light conditions. We then processed the samples through wet sieving and utilized lithium metatungstate
to separate the feldspar grains based on density $\rho$ <2.565 g cm$^{-3}$ (Rhodes, 2015). Luminescence measurements were measured
using a TL-DA-20 Risø automated reader equipped with a single-grain IR laser (830 nm, at 90% of 150 mW) and $^{90}Sr/^{90}Y$ beta
source (Bøtter-Jensen et al., 2003). Emissions were detected using a photomultiplier tube with the IRSL signal passing through
a Schott BG3-BG39 filter combination. Full descriptions of dose rate calculation, post-IR IRSL protocol, and anomalous fading
test can be found in the Supplemental Information.

We estimated equivalent doses for 158 K-feldspar grains (Table 2) and determined single-grain luminescence age populations
following the methods detailed in Saha et al. (2021). The sample show a high degree of overdispersion (OD 42%) between





single-grain De. Such high OD values may indicate incomplete bleaching during their last transport and depositional event (Figure 7a, Table 2, Rhodes, 2011). Following the recommendation of Rhodes (2011, 2015), a conservative OD of 15% is

used as a cutoff between well bleached and partially bleached samples. We therefore used a 3-parameter minimum age model (MAM) after Galbraith et al. (1999) assuming a standard OD (i.e., sigmab) of 15% in R statistical package for luminescence dating (Kreutzer et al., 2012). The post-IR IRSL ages was calculated in the DRAC 1.2 online calculator (Durcan et al., 2015). We assumed that the MAM represents the most recent depositional age of the sand deposits at the gravel quarry (Figure 7).

## 4 Results

### 4.1 Geomorphic mapping

Quaternary geologic and geomorphic mapping resolves multiple stages of incision in the San Juan Canyon between Raplee Ridge and Mexican Hat. We identified multiple eroded fill terraces higher than the terrace sampled for [10]Be cosmogenic radionuclide and p-IR IRSL analyses, as well as multiple lower strath terraces. Figure 4a-c shows the preliminary results of both SfM orthographic imagery and satellite imagery geomorphic mapping. Fluvial deposits can be found on both banks of the

San Juan River; however, straths are typically unpaired in any single reach.

The paired orthographic imagery and DEMs from the SfM surveys allow us to classify fluvial deposits based on the elevation of the strath surfaces on which they rest. We identify a series of at least six terraces with elevations ranging from 10-60 m above the modern San Juan River (Figure 4d). Q1 is found in the upstream portion of the surveyed area and is roughly 10 m above the modern San Juan River. Q2 (∼20 m above modern San Juan River) is found on the southeast side of the San Juan River across

from the Mexican Hat quarry and parallels the modern San Juan River. Q3 (∼32 m above river) and Q4 (∼41 m above river) are the most extensive, paralleling the San Juan River in some locations and occupying the ends of minor valleys orthogonal to the modern San Juan River (Figure 4a). The Q3 fluvial deposits were dated using p-IR IRSL and cosmogenic radionuclide techniques, where they overlie a strath surface 32 m above the modern San Juan River. Two higher but deeply eroded deposits, Q5 (∼55 m above river) and Q6 (∼60 m above) mantle isolated hills in the upstream reach of the survey regions and may

be either lag gravel remnants of fluvial deposits or colluvium. Q3 is the thickest fluvial deposit at ∼3-7 m, while the others are typically several meters but locally can be <1 m. Additionally, we used a combination of field observations and Google Earth to map fluvial material around Mexican Hat. These units are numbered independently and denoted with "Q?" as we lack sufficient data to determine exact elevations or correlate them with our deposits in the SfM area. Some of these deposits are older than Q6, with strath heights on the order of 100 m above the river.

The location and spatial pattern of fluvial deposits within the study area resolve two cutoff bedrock meanders in the direct proximity of our sampling location (Figure 5). Both are now drained by tributary streams. Eroded fluvial deposits are exposed throughout the smaller meander and in parts of the larger one where not buried by younger, locally derived alluvium. Near the Mexican Hat Rock quarry these cutoff meander features separate Q3, which predates the meander cutoff, from Q2, which parallels the modern San Juan River and appears to be the first strath formed after the meander was cutoff. The modern river



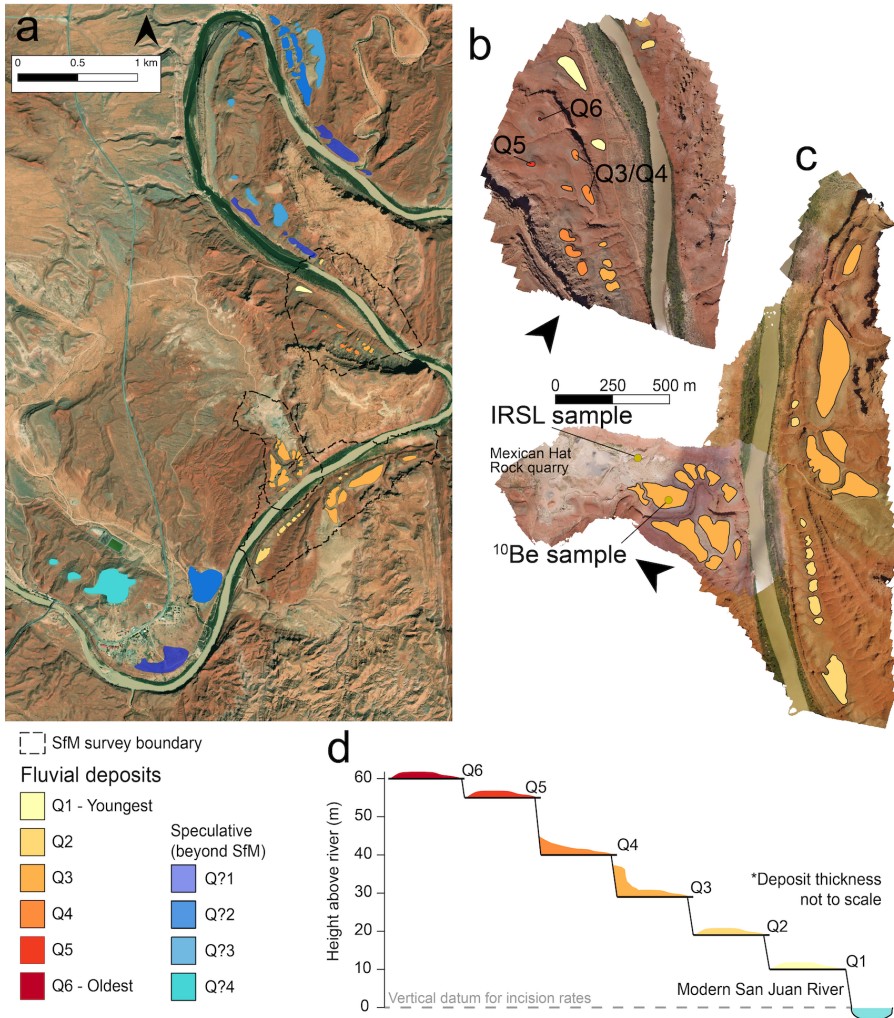

**Figure 4.** a) Satellite imagery of the San Juan Canyon near Mexican Hat with both SfM and satellite-derived geomorphic map overlays. Satellite imagery collected Oct. 4, 2017, Maxar Vivid Imagery, accessed through ESRI World Imagery. b) Detailed geomorphic map of upper canyon created using both field observation and SfM-derived orthographic imagery. c) Detailed geomorphic map of lower canyon and gravel quarry created using both field observation and SfM-derived orthographic imagery. [10]Be and IRSL dating was performed on fluvial terrace Q3. d) Strath surface elevations relative to the modern San Juan River in SfM-derived DEM and orthographic imagery mapping.

in the bypass reach flows south, whereas before cutoff, flow in this reach was toward the north. This may be the only location among major rivers upstream from the Grand Canyon where such a local flow reversal has occurred.

Between Raplee Ridge and the Mexican Hat Rock quarry, the SfM and satellite imagery mapping suggest bedrock channel migration to the west/northwestern, outer bank of the meander, and down-stepping bedrock "scroll bars" along the inside of





the first meander bend in the San Juan River downstream of Raplee Ridge (Figure 5b). There is also a northeast-migrating

down-stepped series of strath surfaces on the opposite bank downstream of this meander.

Farther upstream, there are multiple bedrock meander cutoffs (Figure 5a), particularly where the San Juan River incises into the resistant Pennsylvanian strata (Leopold and Bull, 1979; Harden, 1982, 1990). These meanders display far greater ridge-to-channel relief than those downstream, which likely reflects the difference in erodibility between the resistant Pennsylvanian limestones upstream of Raplee Ridge monocline and the mudstones of Halgaito Tongue Formation downstream of the mon-

ocline. The sinuosity of the deeply entrenched meander cutoffs is similar to those found in the Goosenecks downstream of Mexican Hat, though no abandoned meanders are observed in that stretch of the river.

### 4.2  Reassessing Bluff terrace dating

Our reassessment of the Bluff terrace data yielded $t = 1.63^{+0.08}_{-0.07}$ Ma, $\epsilon = 9.8^{+0.8}_{-0.6}$ m Myr$^{-1}$, $\rho$ = 1.77 g cm$^{-3}$, and the addition of no material to the terrace surface (Figure 6a). Clast inheritance values ranged from 0.112–1.622 × 10$^6$ atm g$^{-1}$ (Figure

6a inset). This increase in age suggests a long-term incision rate of 84–101 m Myr$^{-1}$, a slightly lower incision rate than the original estimate for this location, 110 m Myr$^{-1}$ Our terrace erosion rate results overlap within error with the 14 $\pm$ 4 m Myr$^{-1}$ calculated by Wolkowinsky and Granger (2004) but are lower than previously estimated. An estimated density of 1.77 g cm$^{-3}$ and the addition of no eroded material are both reasonable values. We discuss the accuracy of this incision rate in more detail in Section 5.2.

### 270  4.3  Mexican Hat terrace age dating

Figure 6b shows $^{10}$Be concentrations as a function of depth at the gravel quarry near Mexican Hat, Utah (Table 1), which yielded an exposure age of 27.9 $\pm$ 0.8 ka (R2 = 0.996) with an inherited $^{10}$Be concentration of 0.712 $\pm$ 0.032 × 10$^5$ atm g$^{-1}$, assuming no erosion has taken place since the exposure of the top of the observed deposit (an assumption addressed below). The catchment-averaged inheritance value is within the 0.27–5.11 × 10$^5$ atm g$^{-1}$ range of $^{10}$Be inheritance values from

Wolkowinsky and Granger (2004).

The MAM for the sample yields an equivalent dose (De) of 141.8 $\pm$ 9.9 Gy and a p-IR IRSL age of 39.8 $\pm$ 3.1 ka (Table 2; Figure 7a). High overdispersion (42%) in our sample likely indicates incomplete bleaching from sunlight exposure, a result much more common from fluvial settings (Figure 7a, e.g., Gliganic et al., 2017). Our single-grain analysis suggests that the sands are likely fluvial in nature rather than aeolian as originally assumed in O'Sullivan (1965), though other sands that do

appear reworked by wind can be found in the region. This is further bolstered by the dominantly subhorizontal, <1 mm bedding in the sands (Figure 7b) and absence of high-angle cross bedding that would indicate aeolian deposition. We assume that the 40 ka from MAM represent the most recent depositional age of the fluvial sands while the of older grains represent remobilized fluvial material whose luminescence signal were not reset during transport.

While not in direct agreement, the relative time of deposition of the overlying sands, as measured by p-IR IRSL, and

exposure of the underlying gravel, as measured by $^{10}$Be depth profile, are correctly ordered, with deposition before exposure. This brackets the age of the deposit and eventual meander cutoff. As the top of the sand deposit is ∼4 m higher than the

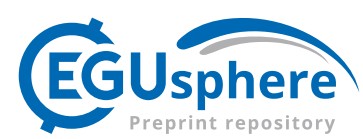

**Table 1.** Cosmogenic radionuclide depth profile data reporting

| Sample | Depth (cm)[a] | Quartz (g) | $^{10}$Be/$^9$Be ratio[b] | $^{10}$Be/$^9$Be ratio uncertainty[c,d] | $^{10}$Be concentration (atm g$^{-1}$)[e] | $^{10}$Be concentration uncertainty (atm g$^{-1}$)[c] |
|---|---|---|---|---|---|---|
| Rap07 | 10 | 15.72 | 2.2518E-13 | 3.74262E-15 | 318980 | 5378 |
| Rap08 | 20 | 104.37 | 1.69545E-12 | 3.17738E-14 | 301557 | 5652 |
| Rap09 | 50 | 113.65 | 1.51039E-12 | 4.06447E-14 | 241981 | 6513 |
| Rap10 | 100 | 74.33 | 7.02632E-13 | 1.14929E-14 | 171817 | 2815 |
| Rap11 | 200 | 80.29 | 4.46936E-13 | 7.33961E-15 | 102001 | 1681 |

Sampling location details and common values between samples: Location (ºN/ºW) = 37.16205/109.84611, elevation (masl) = 1278, thickness (cm) = 2, shielding factor = 1, denudation (mm/yr) = 0, density[f] (g cm$^{-3}$) = 1.77, surface production rate (atm g$^{-1}$ yr$^{-1}$) for spallation = 10.1979 and muon = 0.1781, LSDn scaling scheme, taken from CRONUScalc (http://cronus.cosmogenicnuclides.rocks/2.1/)

[a]Depths measured from the top of gravel pavement surface.

[b]Isotope ratios were standardized to $^{10}$Be standards prepared by Nishiizumi et al. (2007), 07KNSTD33I0, with a value of 2.85 x 10$^{-12}$ and $^{10}$Be half-life of 1.39 x 10$^6$ years.

[c]Uncertainties reported at 1-sigma level.

[d]Propogated uncertainites include error in the blank.

[e]Blank correction was performed by subtracting $^{10}$Be atoms in procedural blank from all samples.

[f]Estimated density for unconsolidated fluvial sediments.



**Table 2.** Dose-rate information and post-IR IRSL ages based on minimum age model (MAM)

| Lab code | Field code | K (%)[a] | Th (ppm)[a] | U (ppm)[a] | Total dose-rate (Gy ka$^{-1}$) | Equivalent dose (Gy) | Corrected post-IR IRSL age (ka) [b, c] |
|---|---|---|---|---|---|---|---|
| SFO19-01 | J1531 | 1.90 | 4.30 | 1.70 | 3.56 ±0.13 | 141.81 ±9.85 | 39.8 ±3.1 |

Sampling location details: Location (°N/°W) = 37.1636/109.8450, elevation (masl) = 1278, depth (m) = 3.2

Grain size used 175-200 $\mu m$.

Radionuclide conversion factor after Liritzis et al. (2013); $\alpha$ attenuation factor after Brennan et al. (1991); $\beta$ attenuation factor after Guérin et al. (2012).

Internal K contents were 12.5±0.5% after Huntley and Baril (1997).

Cosmic dose rates following Prescott and Hutton (1994).

[a] U, Th and K contents derived via ICP-OES with relative uncertainties of 5%.

[b] Ages are calculated in DRAC-calculator (Durcan et al., 2015).

[c] No fading



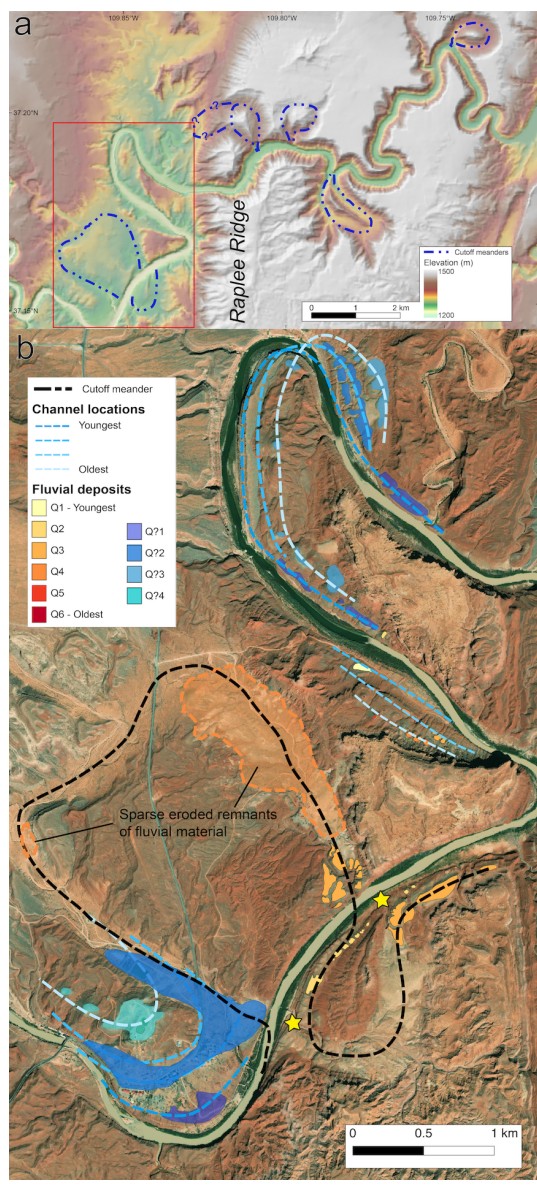

**Figure 5.** a) Shaded-relief map of the Mexican Hat region. Cutoff bedrock meanders shown in blue dashed lines. Red box denotes extent of Figure 5b. Digital elevation model and derived hillshade created using 1 arc-second US Geological Survey National Map 3D Elevation Project (3DEP, https://www.usgs.gov/core-science-systems/ngp/3dep). b) Satellite imagery (collected Oct. 4, 2017, Maxar Vivid Imagery, accessed through ESRI World Imagery) of San Juan Canyon near Mexican Hat, Utah. Mapped fluvial deposits are shown with proposed migrating channel locations (blue lines), San Juan River prior to meander cutoff (black dashed lines), and approximate meander cutoff locations (yellow stars).





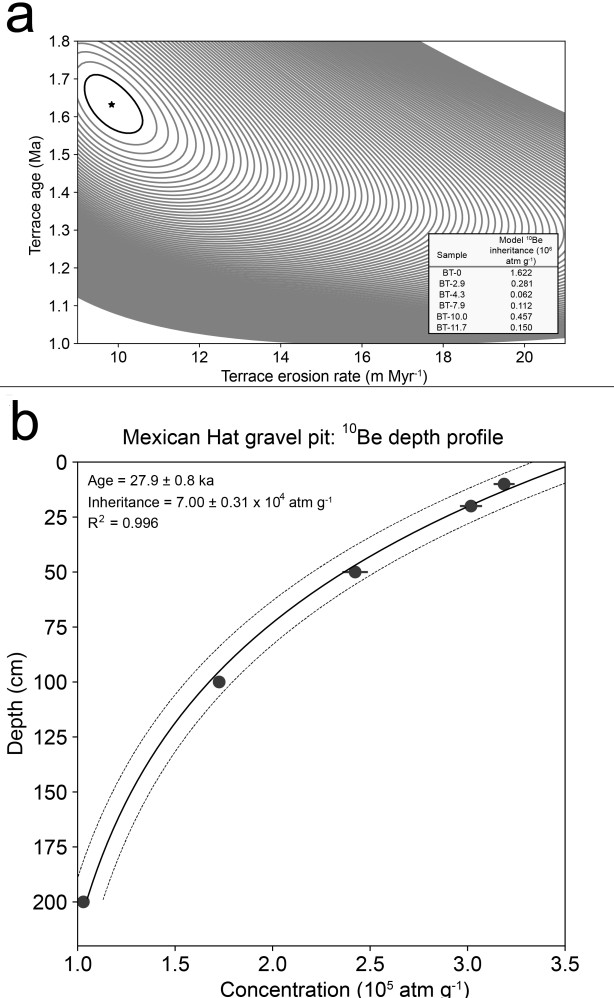

**Figure 6.** a) Recreation of best-fit terrace erosion rate and age from Bluff data (Wolkowinsky and Granger, 2004). Contours are one standard deviation. Star represents best-fit value. b) Depth profile of $^{10}$Be concentrations with error bars. Dotted lines represent 1-sigma error of best fit as determined by nonlinear least squares. Sample taken at 37.16205º N, 109.84611º W, and 1278 m elevation. No surface erosion or shielding factors applied.

top of the gravel pavement surface, we believe the gravels were largely shielded from cosmogenic rays by the sand deposit. After meander cutoff and channel abandonment, erosion would eventually remove the sand and expose the gravels, forming the currently observed pavement surface that serves as the top of the gravel depth profile. We view this scenario as likely and

believe the $^{10}$Be depth profile exposure age likely underestimates the time of the meander cutoff. Nonetheless, the meander cutoff could have occurred no earlier than the age of deposition of the units that were bypassed (∼40 ka), and no later than the exposure age of the gravel package (∼28 ka).





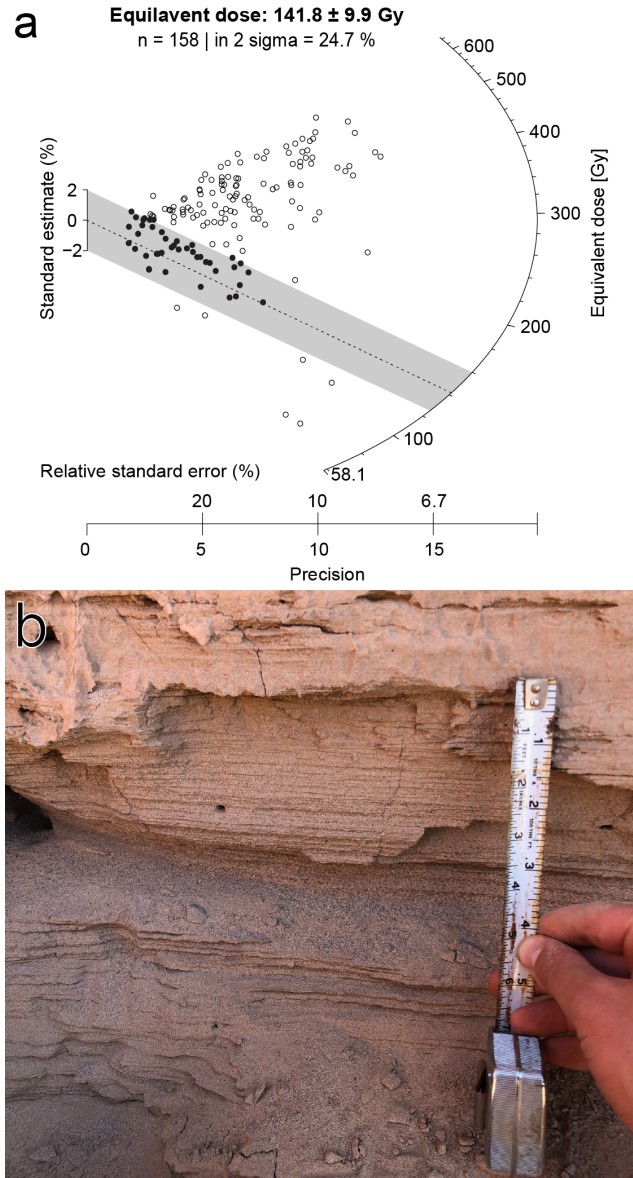

**Figure 7.** a) Radial plot showing single-grain equivalent doses plot showing minimum age model modeling. b) Horizontal, <1 mm scale bedding in sand deposit.

The 32 m strath elevation above the modern San Juan River suggests a short-term bedrock incision rate between 804–1151 m Myr$^{-1}$ over the last ∼28–40 ka, though the higher bound likely overestimates incision as it relies on the gravel exposure age. This range is higher than the incision rate inferred for the San Juan River ∼45 km upstream in Bluff, Utah, particularly given our revision age of the Bluff deposit from ∼1.36 to ∼1.63 Ma adjusts the incision rate to 84–140 m Myr$^{-1}$ This incision rate is



also on the high end of incision rates measured for the San Juan River near Navajo Mountain (425-800 m Myr[-1] Garvin et al., 2005).

## 5 DISCUSSION

### 5.1 Paleochannel geometry, preservation of meander cutoffs, and incision rate

Although meanders are typically being associated with alluvial rivers, active bedrock meanders are well-documented in studies of bedrock channel strath formation (Hovius and Stark, 2001; Finnegan and Dietrich, 2011). Multiple locations within our field area (Figure 5b) show evidence of down-stepping unpaired strath terraces on the inside of bends, typically a sign of bedrock meandering (Personius et al., 1993; Finnegan and Dietrich, 2011). As noted by Finnegan and Dietrich (2011), bedrock meander cutoffs result in knickpoints that can propagate upstream, which would enhance local vertical incision.

Variations in bedrock hardness have affected the relief between the channel and surrounding uplands along the river. Cutoff bedrock meanders with floors in the erodible Halgaito Tongue are typically entrenched ∼40 m (but up to 100 m) below surrounding mesas in the Cedar Mesa sandstone. In contrast, those with floors in the Hermosa Formation are typically ∼200 m below plateaus formed by the resistant Rico Formation. Also, the active channel and better-preserved cutoff meanders in the Rico/Hermosa have rather symmetric cross-valley profiles, whereas some in the Halgaito Tongue/Cedar Mesa are markedly asymmetric. Given that fluvial bedrock erosion consists of both vertical and lateral erosive processes, these topographic differences imply a higher rate of vertical relative to lateral erosion in resistant layers, allowing considerable deepening prior to meander cutoff. Conversely, meanders in erosive lithologies appear to incise less before lateral migration causes a cutoff in the meander. This implies a higher frequency of meander cutoffs in erodible lithologies compared to resistant lithologies, resulting in higher variability in measured incision rates depending on the spatial and temporal proximity to cutoff events. It is plausible that bedrock meanders upstream from Raplee Ridge deepened primarily through vertical incision while in resistant limestones of the Rico Formation and that lateral migration and cutoff occurred when the river encountered more erodible shale layers in the underlying Hermosa Formation. This would require higher-resolution elevation data and field observation to confirm.

To determine whether meander cutoff can explain the observed discrepancy in short- versus long-term incision rates observed along the San Juan River, we constructed a simple geometric model in which the instantaneous cutoff of a meander strands sediments residing at the upstream end of the meander. In this construction, the preferred wavelength of an initially straight channel is denoted as L, and evolves to a particular sinuosity, $S$, at which time the channel bypasses the meander (Figure 8a). We further assume the slope of the channel outside the meander ($S_c$) remains approximately constant during meander formation (Sieh and Jahns, 1984). Finally, we assume that the length of the bypassing segment is small relative to $L$.

With these assumptions, the stream-wise length of the channel segment above the meander cutoff ($l_d$) is

$$l_d = L\left(S - \frac{1}{2}\right) \tag{3}$$

which implies an elevation difference across the meander at the moment prior to cutoff of

$$\Delta h = S_c L\left(S - 1\right) \tag{4}$$





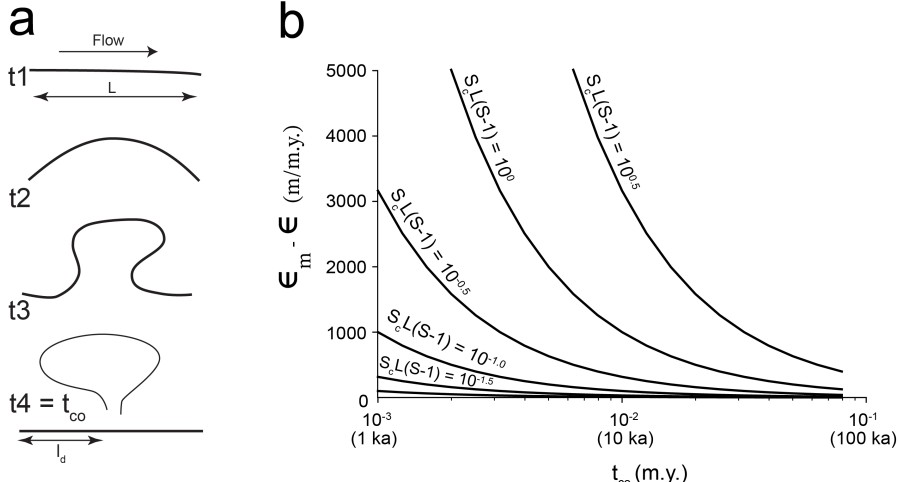

**Figure 8.** a) Schematic diagram of meander cutoff and evolution through time with length of straight channel ($L$) and channel segment upstream of cutoff ($l_d$). b) Plot of time since meander cutoff ($t_{co}$) versus the difference between long- and short-term incision rates ($\epsilon$ and $\epsilon_m$, respectively) for various channel slope ($S_c$) and sinuosity ($S$) combinations.

In this case, $\Delta h$ represents the elevation difference that results solely from the bypass of the meander bend. From the time of the cutoff until the present $t_{co}$, the total accumulated elevation between the channel bottom and the top of the deposit is:

$$\Delta h + \epsilon t_{co}. \tag{5}$$

Thus, the measured short-term incision rate ($\epsilon_m$) inferred from the age of the deposit abandoned by cutoff and the current river elevation is

$$\epsilon_m = \frac{\Delta h}{t_{co}} + \epsilon \tag{6}$$

such that the difference between the short- and long-term incision rates will be:

$$\epsilon_m - \epsilon = \frac{\Delta h}{t_{co}} = \frac{S_c L (S-1)}{t_{co}} \tag{7}$$

When the cutoff has happened relatively recently ($t_{co}$ is small), the short-term incision rate will be strongly influenced by the cutoff process, whereas as time passes ($t_{co}$ is large), $\epsilon_m$ will approach $\epsilon$. The magnitude of the discrepancy at a given value of $t_{co}$ will vary in proportion to the amount of relief created by the cutoff event ($S_c L (S-1)$, Figure 8b). For small cutoff meanders with low relief from the upstream to downstream end, the discrepancy in erosion rates will only be appreciable for a short time period, while large cutoff meanders with significant elevation difference across the ends may impact erosion rate discrepancies for 10-100 ka.

Importantly, each of the terms in Equation 7 can be measured or inferred based on field observations, which allows evaluation of the effect of meander growth and cutoff to be assessed at particular sites. At Mexican Hat, the current value of $S_c$ between





Comb Ridge and Mexican Hat is $1.2 \times 10^{-3}$ based on measurements from the Cortez 1:24,000 Quadrangle (the San Juan River undergoes an elevation drop of 48.8 m over 40.5 river kilometers). Likewise, we approximate the initial meander wavelength of the San Juan in this area to be around 1.2 km based on the current channel geometry and the cutoff length observed outside of Mexican Hat. This being the case, the length of the two mapped, cutoff meanders in Figure 5B (∼9.1 km) implies a sinuosity of around 7.6. Finally, we use the $^{10}$Be exposure age dating (∼28 ka) to approximate the timing of cutoff of the meander ($t_{co}$).

Using these values, we would expect a difference between the incision rate since the meander was cutoff and the long-term incision rate to be ∼342 m Myr$^{-1}$ Using the measured short term incision rate (804-1151 m Myr$^{-1}$) and the range of measured long-term incision rates (84-140 m Myr$^{-1}$, depending on dating method), our calculations yield only ∼32–52% of the observed discrepancy between short- and long-term incision rates. While appreciable, this effect cannot entirely reconcile the observed divergence in incision rates.

Late Cenozoic climatic cycles are known to result in fluvial terraces in both glaciated and unglaciated catchments worldwide (Westaway et al., 2009). The cycles of incision and aggradation observed both prior to and after meander cutoff near Mexican Hat suggest that other exogenous effects, plausibly glacial-interglacial cycling, modulate the patterns of alluviation and thus incision rate over time. This phenomenon has been specifically invoked to explain the long-term terrace record in the broader Colorado Plateau (e.g., Bridgland and Westaway, 2008). The San Juan Mountains have an extensive history of glaciation

(Atwood and Mather, 1912; Johnson et al., 2017), and so sediment supply and alluviation in the San Juan River must have been impacted by glacial advance and retreat in the headwaters. The Rocky Mountains lack absolute age constraints for the onset of the last glaciation, making it difficult to directly relate the Mexican Hat fluvial deposit to a particular glaciation, nor do our ages clearly align with any marine isotope stage typically associated with glaciation. Nonetheless, there have been a series of documented landscape adjustments in the San Juan Mountains (Johnson et al., 2011) that occurred after the

documented meander cutoff. Such adjustments may reflect variations in runoff, sediment supply, and sediment caliber that might also enhance incision rates downstream. Thus, a combination of local autocyclic processes and regional changes in sediment delivery and runoff to the area together might explain the difference between short- and long-term incision rates along the San Juan River at the Mexican Hat site.

## 5.2  San Juan River incision rates relative to the Colorado River and tributaries

The short-term incision rate near Mexican Hat is significantly higher than estimates for the San Juan River near Bluff and is closer to the range observed near Navajo Mountain at the confluence of the San Juan and Colorado Rivers (400–825 m Myr$^{-1}$). By combining dating methods that record deposition (p-IR IRSL) and exposure (in-situ cosmogenic $^{10}$Be depth profile), the timing of channel abandonment at the quarry must be between ∼27.9–39.8 ka, representing an incision rate of 804–1151 m Myr$^{-1}$ This is nearly an order of magnitude higher than the cosmogenic burial isochron ages at the Bluff site, originally

calculated as 110 m Myr$^{-1}$, and here reassessed to be 84–101 m Myr$^{-1}$ Detrital sandine dating by Heizler et al. (2021) shows a maximum depositional age for the same deposit of 1.208 Ma. As the deposit cannot be younger than the grains that comprise it, we take this to be the most robust age. We suggest the older cosmogenic burial isochron age may be the result of more complex exposure and burial history (as suggested by Hanks and Finkel, 2005) or may represent the onset of deposition rather





than terrace abandonment (as noted by original authors in their reply to Hanks and Finkel). We take the maximum depositional

age from detrital sandine to calculate the minimum long-term incision rate on the San Juan River in this area as 140 m Myr$^{-1}$, closer to our short-term incision rate but still slower by at least a factor of five.

Our incision rate estimate falls on the upper end of 250–725 m Myr$^{-1}$ short-term incision rates for the upper Colorado River and its tributaries (Darling et al., 2012; Aslan et al., 2019). As noted above, there are multiple potential explanations for the observed differences in incision rate: methodological biases from exposure age dating, spatial or temporal fluctuations

in erosion rate due to drainage integration or tectonism, and biases toward higher incision rates when measuring over short timescales (Finnegan et al., 2014). Given the combination of dating techniques used in this work, we view the first of these (methodological biases) as an unlikely explanation for the discrepancy, at least at the Mexican Hat site.

The greater Colorado River system appears to have had changes in incision rates that may be related to tectonism or drainage reorganization. Short-term incision rates in the Glen Canyon area show an acceleration of incision rate from ∼120 m Myr$^{-1}$

to ∼300–900 m Myr$^{-1}$ over the past ∼250–500 ka (Garvin et al., 2005; Cook et al., 2009; Darling et al., 2012), with this acceleration being attributed to the combination of an upstream-propagating incisional wave resulting from Grand Canyon integration and the increased erodibility of the Glen Canyon Group sandstone immediately downstream of Lees Ferry. Darling et al. (2012) specifically noted the uncertainty of the state of such an incisional wave on the San Juan River. They highlighted that while the Bluff terrace (150 m above the modern channel) measured by Wolkowinsky and Granger (2004) correspond in

elevation with those of terraces at Bullfrog near Glen Canyon, long-term incision rates at Bluff were lower than at Bullfrog (116–138 m Myr$^{-1}$, Darling et al., 2012). However, the detrital sandine dating at Bluff brings these incision rates into alignment, suggesting an incisional wave may have reached this stretch of the San Juan River and be contributing to elevated short-term incision rates.

Simple detachment-limited bedrock channel incision modeling has shown that lithologic contrasts may enhance upstream

propagation of such an incisional wave once it encounters more erodible strata, increasing short-term incision rates (Cook et al., 2009; Darling et al., 2012). Cook et al. (2009) documented such accelerated incision rates in Trachyte Creek, a minor tributary of the Colorado River that enters Glen Canyon roughly ∼135 km upstream of the Colorado's confluence with the San Juan River. Given the similarity of soft-over-hard lithologic contrast between Paleozoic limestones and Mesozoic sandstones along the San Juan River upstream of the confluence with the Colorado River, it is reasonable to assume a knickpoint would

propagate similarly due to equivalent lithologic contrasts.

Differences in short-term and long-term incision rates have been observed elsewhere on the Colorado Plateau (Aslan et al., 2019). In the upper Colorado River basin, short-term incision rates show significant variation, even within individual regions, while long-term incision rates are both lower and are more consistent with one another (Aslan et al., 2019). Aslan et al. (2019) invoked tectonic and isostatic uplift of the flanks of the Rocky Mountains to explain the spatial and temporal incision rate trends

observed. However, in the absence of other evidence for such uplift in the Mexican Hat area, we conclude that this explanation would be insufficient to explain our increased incision rates.

This leaves other factors to produce higher incision rates measured over shorter timescales. Finnegan et al. (2014) noted the sensitivity of incision rates to measurement interval, citing the non-steady-state nature of rivers experiencing alluvial



aggradation resulting from factors including climate and sediment supply. Additionally, Aslan et al. (2019) pointed to local factors such as bedrock lithology, glacial history, and drainage reorganization as modulating incision rates, with the largest incision rates exceeding >3000 m Myr⁻¹ on the Gunnison River. Meander cutoffs provide a novel mechanism by which this "Sadler effect" may occur. Given that the sampled deposits are within a cutoff bedrock meander and proximal to a second such meander, enhanced local incision is likely at least partially responsible for the almost order-of-magnitude increase we observed at Mexican Hat relative to the long-term rate at the Bluff site. As noted above, the process of meander growth and eventual cutoff may increase short-term incision rates relative to their long-term average, as the shortened channel length after the cutoff may juxtapose sediments deposited on the upstream end of a meander with the bypassed segment of the meander. Our work supports meander cutoff effects accounting for up to 50% of incision rate discrepancy near Mexican Hat, however the remainder may be attributable to combinations of top-down (climatic/sedimentological) and bottom up (drainage integration/tectonic) processes.

## 5.3  Broader implications

Gilbert (1877) first described the connection between the amount of alluviation and a river's ability to vertically incise, coining the term "planation" to describe the lateral carving of broad plains when vertical incision is slow. Specifically, he noted that "…downward wear ceases when the load equals the capacity for transportation. Whenever the load reduces the downward corrasion to little or nothing, lateral corrasion becomes relatively and actually of importance" (p. 126). Recent advances in mechanistic modeling of mixed bedrock-alluvial systems demonstrated this same point, showing that rates of bedrock meander development are significantly affected by patterns of alluviation (Inoue et al., 2017, 2021). These studies found that increases to initial alluvium thickness decreased vertical erosion while increasing the rates of lateral erosion. Applying this to a river such as the San Juan with headwaters that have experienced glaciation suggests that the availability of alluvium due to glacial-interglacial cycling should impact meander migration rates. Rock tensile strength (one potential measure of erodibility) impacts rates of both vertical and lateral erosion in the models of Inoue et al. (2021). However, vertical erosion is sensitive to the areal fraction of alluvial cover on the channel bed while lateral erosion rates are insensitive to the ratio of cover. Under alluvial deposition, lateral migration would be expected to continue even if vertical incision is significantly impeded by alluviation, however rates would vary significantly depending on the lithologic susceptibility of the bank to erosion, whether by abrasion or plucking. Gilbert (1877) specifically noted the patterns of alluviation and planation in soft rocks along Trachyte Creek, showing that maintenance of a constant river profile resulted in transport-limited conditions in erodible lithologies, creating higher lateral channel migration.

Notably, the high rates of channel migration in an erodible lithology are more likely to result in stranded, dateable geologic material than in a resistant lithology. Increased alluviation in a meandering reach increases the frequency of cutoffs, which strands material suitable for cosmogenic or luminescence dating. This biases these dating techniques towards periods of punctuated changes to local vertical incision rates. Given alluviation impedes vertical incision rates and increases channel migration (leading to higher rates of channel abandonment that preserves datable material) lithologic context are important for interpreting vertical incision rates. This is particularly important over timescales that are similar to or less than the mean recurrence of punctuated events like meander cutoff.





When locations with preserved fluvial material are sparse, interpretations of regional incision rates necessarily rely heavily on a handful of sampling localities, some of which may be particularly sensitive to local effects resulting from variations in lithology and the planform evolution of channels. This becomes increasingly true in estimates of short-term incision rate, where the temporal scale is insufficient to average over repeated punctuated episodes of incision resulting from autogenic processes and channel dynamics. When attempting to connect widely separated remnants to understand the current state of a fluvial system, interpretation gets even more fraught. While far-felt effects of drainage integration will impact incision rates, this signal may be convoluted with these shorter timescale and local effects.

## 6 Conclusions

Fluvial deposits in the San Juan Canyon near Mexican Hat, Utah dated using p-IR IRSL and cosmogenic radionuclide [10]Be depth profile methods suggest that the San Juan River has experienced a short-term bedrock incision rate of 804–1151 m Myr[-1] over the past ∼28–40 ka. This rate is within the range of other short-term incision rates throughout the Colorado Plateau. While increases in incision rates elsewhere are attributed to the upstream propagation of knickpoints due to base level fall as the Colorado River integrated through Grand Canyon and downstream basins, high short-term incision rates in the San Juan Canyon likely reflect, at least in part, cutoff bedrock meanders in the area surrounding the sample site. Evidence for additional incision and aggradation cycles following meander cutoffs in the San Juan Canyon may be correlated with the changing sediment supply during glaciation in the mountainous headwaters of the San Juan River. Preservation of fluvial sediment may be biased by planform rearrangements that also impact short-term vertical incision rates, requiring enhanced scrutiny of the geologic and geomorphic context of the few deposits remaining in a largely erosional landscape, such as the Colorado Plateau.

*Data availability.* All data reported in manuscript is available as spreadsheets via (will insert Zenodo repository DOI upon finalization of publication details).

*Author contributions.* Hilley, Seixas, and Steelquist conceptualized the project. Steelquist, Seixas, and Gillam performed field work and sample collection. Steelquist, Seixas, Saha, and Moon performed laboratory work. Steelquist and Gillam performed digital mapping. Hilley developed conceptual model for cutoff impacts on incision rate. Steelquist and Hilley wrote initial manuscript and all authors contributed to editing and finalization.

*Competing interests.* The author has declared that there are no competing interests.



*Acknowledgements.* Thanks to S. Johnstone, H. Young, and C. Baden for assistance collecting the $^{10}$Be samples from the Mexican Hat quarry in addition to drone surveys in the area. Thank you to Billy and Brooke Gaines for access to the Mexican Hat gravel quarry. Special
thanks to the Navajo Nation for allowing us to perform aerial survey and reconnaissance fieldwork on the south side of the river, in particular Akhtar Zaman, Rowena Cheromiah, Richard Carlton, and Steven Prince at the Navajo Nation Minerals Department. This work was funded through Stanford McGee/Levorsen grants to Steelquist.



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
