# Peer review of "The impact of bedrock meander cutoffs on 50 ka-year-scale incision rates, San Juan River, Utah"

_EGUsphere, 2024_

## Author Comment (AC1)

Author comments in red
Reviewer comments in black

We thank the editor and both reviewers for their careful and considerate assessment of our work. The revised manuscript is markedly improved, in our opinion, for having incorporated the variety of clarifications and revisions suggested.

Though not related to any specific reviewer comment, we wanted to be transparent that during the revision process, a reassessment of the strath terrace elevation used to calculate our incision rate resulted in the elevation of the strath going from 32 m to 27 m. M. L. Gillam was in the field and did a more detailed measurement of strath elevation and found that the location where the SfM-derived elevation was measured is anomalously high compared to the majority of the strath surface, particularly near the gravel quarry sampling site. Our results are still consistent with the original manuscript, but the revised incision rates are now 678-968 m Myr$^{-1}$ instead of 804–1151 m Myr$^{-1}$.

Reviewer #1

This manuscript presented river incision rates of the Mexican Hat reach of San Juan river, inferred from dating of exposure and deposition age of the terrace fill. The inferred incision rate which averages over ~50 Ka is one order of magnitude higher than the incision rate inferred from a ~1.2 Ma aged terrace fill of the Bluff terrace upstream of the Mexican Hat reach. The incision rate difference between the two terrace sites is hypothesized to be caused by a meander-cutoff event at the Mexican Hat. A quantification model of cutoff-induced erosion intensification is proposed and applied to the Mexican Hat reach, revealing that 50% of the incision rate difference can be explained by the cutoff. The authors also discussed other alternative mechanisms, including base-level change erosion wave propagation of the bigger area, glaciation of the source area, and lithological effects. I think this manuscript is generally well-drafted, the presentation is in reasonable logic flow, and the meander cutoff model is interesting. Before this manuscript is advancing to the next phase, I think a few questions need to be addressed:

**Specific comments :**

- The causal relationship between the terrace Q3 and the meander cutoff is not evidenced clearly. Meandering cutoff age is constrained by the dated terrace at the quarry, Q3, in this study. I think it needs to be justified with more details. At the quarry, the meander course cuts through the Q3 according to the descriptions in section 4.1. That is saying the Q3 is formed before the cutoff, not formed from the cutoff event. The relative time between Q3 exposure and cutoff is unknown.

  We apologize for the confusion here. The reviewer seems to have the impression that we think that the river continued to flow around the large, now-abandoned meander after Q3 was exposed. Instead, we believe that Q3 was exposed when the meander was abandoned due to cutoff. There are excellent exposures of river deposits within the

gravel pit and southeastward toward the river but no inset terrace deposits younger than Q3 suggest the river continued at a lower elevation within the meander NW of the modern San Juan River prior to abandonment. The contact between the strath and the terrace fill is subhorizontal along the entire length of its exposure towards the modern San Juan River canyon. Given these lines of evidence, we feel confident in our interpretation that the large meander (including the Q3 material within it) was abandoned at the time of cutoff.

To clarify this point for readers, we have described this site in more detail in Section 4.1 and explained our thinking more clearly: "Near the Mexican Hat Rock quarry these cutoff meander features contain Q3, suggesting Q3 predates the meander cutoff. Within this cutoff meander, there is no inset strath terrace, and its southeast edge has excellent exposure of the contact between the Q3 strath and its fill. The strath is subhorizontal and has some minor undulations, but we did not observe any evidence of subsequent incision after deposition of the fill, though gullying crosscuts both fill and strath on the southwest edge of the gravel pit. The major strath surface below is Q2, which parallels the modern San Juan River in this area and predominantly appears on the opposite bank of the river from the gravel quarry (Figure 5a). We interpret this to be the first strath formed after the meander(s) cutoff. Given this, we interpret that the meander cutoff resulted in the abandonment of Q3. The modern river in the bypass reach flows south, whereas, before the cutoff, flow in this reach was toward the north. This is the only documented location along the San Juan River where such a local flow reversal has occurred."

Two patch clusters of Q3 are also found upstream of the quarry and the meander cutoff (Figure 5b, colored with orange and blue). The elevation difference between Q3 strath and the modern San Juan river bed is not 100% from the intensified erosion due to the cutoff. Should the relief difference (relative to the modern San Juan bed) between the Q3 patch clusters that were affected by the cutoff and those not affected by the cutoff be used to justify the intensified erosion due to the cutoff? I wonder what the authors think about this problem?

The mapped Q3 patches upstream of the quarry have a strath height very similar (within ~2 meters) to the Q3 patch mapped at the quarry. We believe the knickpoint resulting from cutoff has subsequently migrated upstream beyond our high-resolution surveys, meaning all mapped terraces predating it have undergone the same pulse of intensified erosion. The bedrock in which the knickpoint would have formed is the Halgaito Tongue Formation, which is predominantly siltstone and thus likely to erode fairly rapidly under a newly formed knickpoint that could migrate upstream.

While not sufficiently studied to go into the main text of the manuscript, there are multiple small sections of rapids ~10 km upstream of the proposed cutoff meander location. This would require a ~0.33 m/yr migration rate which isn't unheard of (Loget and Van Den Driessche, 2009).

This comment also highlighted a confusion that the "Q?_" units correspond to mapped Q1-Q6 units. The elevations of Q?_ units are poorly constrained, thus we simply wanted to draw attention to their existence rather than attempt to correlate them with others in the region. Specifically, they are not meant to correlate with the numerical Q1-Q6 units. To make this clearer, we have amended these coarsely mapped units as follows: "These units are denoted with "Q?" and use ordered letters moving up from the river ("A" is lowest elevation, "B" is next lowest elevation, etc.)..."

In addition, Figure 4c doesn't give enough topographic resolution to show the cut-cross relationship between the Q3 patch and the paleo- meandering river course. The paleo-meandering river course (black dash line) is drawn seemingly randomly on Figure 5b, instead of demonstrated by a high-resolution topographic map in the background, or by field evidences.

We have updated the map in Figure 5a (previously 5b) to the highest available topographic resolution to better show the relief structures that are evidence for the paleo-meander. This also more clearly highlights abandoned fluvial material found throughout the meander (far left of Figure 5a). We have also expanded the description of our field evidence that supports our interpretation of the meander cutoff (see response to the first "Specific comment" above).

- Section 3.2 and 3.3
- In the presentation of section 3.2, sampling strategy, sample type, and sampling locations are missing in this section. Although the authors used published data of Wolkowinsky and Granger (2004), it is necessary to briefly introduce the sample strategies and samples that have been re-analyzed in this study.

We have updated this section to include a summary of the sampling strategy and type along with a reference to the figure with sample location: "previous work at the Bluff site (Figure 1) exploited both the near-surface accumulation of 10Be and 26Al, as well as the more rapid decay of 26Al at depth to jointly solve for the exposure age, erosion, and inheritance of clasts (Wolkowinsky and Granger, 2004). These researchers collected seven clast samples from terrace fill gravels >13 m thick, with samples taken between the surface and 11.7 m depth. In some cases, these samples were amalgamations of 20-30 quartzite clasts. However, since the time of that initial study, the production and decay rates..."

And in addition, locations of these re-analyzed samples from Wolkowinsky and Granger (2004) indicated on Figure 1 should be indicated distinctively to stand out from the background.

We have changed the symbol to red and highlighted it in the figure caption to make this location clearer.

- I think section 3.2 and section 3.3 should be merged into one section. The methods of these two sections are the same, but apply two (same-type of samples I assume) different terrace sites. I would recommend the author to structure the new section 3.2 into theories and supporting equations, then present the Mexican Hat sampling strategies and samples, the Bluff re-analyzed samples come to the end.

  While we understand the reviewer's interest in combining these two sections as they are both cosmogenic isotope methods, we believe there are sufficient differences in both aim and method to warrant keeping them separated.

  The aim of section 3.2 is to focus on the reassessment of existing data with new constants. These data rely on two isotopes (10Be and 26Al) and are used to simultaneously solve for terrace exposure age, terrace erosion rate, and deposit density. Additionally, these samples rely on individual (or amalgamated) fluvial gravel clasts. In contrast, the depth profile method covered in section 3.3 relies only on 10Be, does not solve for density or erosion rate (only terrace exposure age), and is based on sand taken from between the gravel clasts. Some of the terms required to solve this equation are taken from the reassessment of the Bluff data. While these two sections rely on the same physical and mathematical theory (hence its introduction in the first of the two sections), we believe keeping them separate highlights the distinction between the reassessment and novel age data.

- Section 4.1

Cutoffs and paleo-meandering course, in addition to the mapped terraces, are presented in this section to assist the demonstration of relative timings of multiple events (terraces formation and two cutoff events). This section is important to understand the purpose and logics of the study of this manuscript. I would recommend some restructure effort to be done for this section. I think the terraces, meandering cutoffs, cut-cross relationship between mapped terrace and paleo- meandering course of the studied San Juan reach and the nearby upstream reach should be presented in this section to frame the problem, and address the scientific problem above (Figure 5a). And it should be expanded to justify the relative timing of cutoff and terrace formation of Q3, and Q2.

We have expanded Section 4.1 and updated Figure 5a (previously 5b) to better demonstrate our interpretations here. See specific comments above for the exact changes.

Actually, this section reveals a bigger problem I see about the presenting style of this manuscript. I think 1) the derivation of the hypothesis needs to be sharpened in the introduction section; 2) backgrounds, methods, and results sections are not phrased to serve for evaluating the hypothesis. They were phrased in a way that is not solving problem-centered which have brought difficulties for reviewing.

We have attempted to clarify the hypothesis and aims of this research in the introduction as follows: "This presents an opportunity to constrain the incision history of the San Juan River over different periods of time at two proximal sites, thus we seek to understand 1) if the San

Juan River short-term (<500 ka timescale) and long-term (>500 ka timescale) are in agreement and, 2) what mechanisms may result in disagreement."

We believe this clarification of our aims to 1) compare short- and long-term incision rates and, 2) understand the mechanics that explain the apparent discrepancy then allows the remainder of the manuscript to be understood more clearly with the additional changes made in response to all comments.

For the second point, I suggest replacing the technology-oriented subtitles of the results sections into subject-oriented subtitles. For example, the "4.2 Reassessing Bluff terrace dating" can be replaced with "4.2 Long-term incision rate of the San Juan River" to match the "long-term" and "short-term" incision rates that were described in the hypothesis.

While we understand the reviewer's inclination, we believe the results section is best segmented into specific methodologies and the objective findings of each. The Discussion section, where these pieces of evidence are brought together to address the hypotheses, has subheadings that reflect how the data sum to address the hypotheses. The complexity of the findings requires the mechanistic hypothesis to be addressed first, as it contextualizes the short-term incision rates, thus Section 5.1, which we have amended to be called "Paleochannel geometry, preservation of meander cutoffs, and short-term incision rate." The long- and (now contextualized) short-term incision rates of the San Juan River are then placed into the broader Colorado River system in Section 5.2.

**Technical correction suggestions:**

- Figure 2, the "T" symbols next to numbers are not explained in the caption or in the figure legend. I guess they are strike and dip of layered outcrops. I would also recommend the authors add explicit descriptions of rock types to the geological units in the legend.

  We have added the strike and dip symbol to the legend as well as abridged geologic descriptions from the source geologic map to each lithologic unit.

- Line 111, the "Cutler Formation" is mentioned only here throughout the manuscript. Seems it is the formation above the Halgaito Tongue Formation. Could you clarify it in the legend of the geological unit of Figure 2?

  Thanks for flagging this, we have changed the language to reflect that we are specifically talking about the Cedar Mesa Sandstone, which is the lowest member of the Cutler Formation.

- Line 153-155, not sure I understand the "long-term" and "our short-term" here. What does it mean? New community-accepted parameter values were used for recalculation, but the exposure or erosion rate with these new parameter values are still the same time scale as that calculated with "old" parameter values. I am confused.

Updated to clarify as follows: "...using modern parameter values to ensure consistency between the estimation of a long-term incision rate at Bluff and our estimation of a short-term incision rate at nearby Mexican Hat (see Section 3.3)."

- Line 170-171, why?

  This was based on a slight misreading of Nishiizumi 2007, where the 10Be half-life revision occurred. We have adjusted the text to better reflect that the correction factor is based on their re-assessment of the 10Be/9Be ratio in the NIST SRM4325 standard against which the Wolkowinsky and Granger 10Be measurements were corrected.

- Line 237-238 and Figure 4c, the IRSL samples are from Q3, but why the Q3 terrace extent in Figure 4c doesn't cover the IRSL sampling location? Is the Q3 too disturbed in the quarry to map a confident boundary of Q3?

  Yes, we opted not to attempt to draw boundaries in the quarry as the majority is significantly disturbed. However, we added a mapped section that extends from the location of the IRSL sample as that region is sufficiently undisturbed.

- Line 245-246: better to indicate the sample locations on Figure 5b again to clarify which two cutoffs because Figure 5a has more than two cutoffs there. Then Figure 5b should be referred to in this sentence, not the whole Figure 5.

  Updated to cite only Figure 5b and updated caption to specify that the two bedrock meanders in Figure 5b are those from this study.

- Line 246: "Both are now drained by tributary streams", I am afraid I can't find the tributaries in Figure 5.

  Removed sentence as no longer relevant to the manuscript.

- Line 248-249: I feel it is a confusing sentence. This long sentence seems to be describing Figure 5b. Could you separate this sentence into two so that one describes the relative spatial relationships, and the other describes the inferred sequential events? And refer to the respective figure in the description.

  This section has been significantly rewritten to better explain the observations in the field that led to our interpretation of Q3 as abandoned due to the meander cutoff. See the comment above.

- Line 263: it will be better to explicitly point out the physical meanings in addition to the symbols only. For example, t, is the exposure age (?), and the mean erosion rate, eta, is the average between t to modern (?).

  Rewritten to describe what each variable is before giving the best-fit value.

- Figure 6: could you indicate the two sites consistently of the a and b in the captions?

  The exact sample location in lat/long and elevation are not found in the original manuscript. Rather than estimate them from Google Earth imagery, we have opted to point readers to the original source to avoid false confidence in the sampling locale.

- Line 322, 323: S and Sc indicate two different physical quantities, better to use more distinctive symbols.

  To avoid any confusion, we have changed sinuosity to "ζ." We are happy to pick a different symbol if the reviewer or editor has a preference for an alternate, but everything we looked at has sinuosity as "S" and that is most commonly used for slope in our experience.

- Figure 7a: legend of dots and circles is missing.

  Updated caption to denote both styles of circles.

- Line 293: I would add "of Q3" to this sentence since there were 6 terraces presented in the "Geomorphic Mapping" section.

  Updated as suggested.

- Line 296: "…to 84–140 m Myr$^{-1}$ This incision rate …"◊ "…to 84–140 m Myr$^{-1}$. This incision rate…"

  Fixed.

- Line 306-309: are you comparing similar-aged cutoffs in different rock types so that the relief difference is from rock-erodibility? Besides, where are they on figures (Figure 5a?)?

  Unfortunately, we do not have age constraints on the other observed cutoff bedrock meanders to be able to definitively answer the question here. The subsequent sentences in the paragraph explain our qualitative observation that deeper meanders would support a higher rate of vertical relative to lateral erosion in more resistant, which has implications for the frequency that vertical incision rates would be perturbed by cutoff events.

  Added language to point to examples in Figure 5a as follows: "Cutoff bedrock meanders with floors in the erodible Halgaito Tongue are typically entrenched ~40 m (but up to 100 m) below surrounding mesas in the Cedar Mesa sandstone, the cutoff meanders of this study being a primary example. In contrast, those with floors in the Hermosa Formation (see cutoff bedrock meanders upstream of Raplee Ridge in Figure 5a) are typically ~200 m below plateaus formed by the resistant Rico Formation."

- Line 314: very interesting implications, is there any other literature to support it?

  We are not aware of other literature that has brought together observations of bedrock meander depth in differently erodible lithologies and the lateral migration rate of bedrock meanders to suggest there would be a connection between erodibility and cutoff rate. Happy to include references to this literature if it exists, otherwise we propose this as a discussion point for the community to consider going forward. Additionally, the final sentence details future work that could better explore this question.

- Line 348: Figure 5B◊Figure 5b

  Fixed

- Line 351: "...rate to be ~342 m Myr$^{-1}$ Using the measured..."◊"...rate to be ~342 m Myr$^{-1}$. Using the measured...". More places miss a period symbol (e.g. Line 374, 375). Please check that carefully.

  Apologies for this, seems to have been a "Find and replace" mistake when formatting "m Myr$^{-1}$" in LaTex. We believe they are all fixed now.

- Line 376: "...depositional age for the same deposit of...", after some literature digging, I realized the "same deposit" is actually the Bluff terrace fill. It does no harm to point it out clearly instead of saying "the same deposit".

  Thanks, we have updated to state clearly that this is the Bluff terrace fill.

Reviewer #2

I think the manuscript by Steelquist et al. is clearly and well written, presents new age dates that will be useful for researchers understanding the evolution of the Colorado Plateau, and should be published after minor revisions. I only have a small number of suggested improvements.

Line 24: add "incision" after Grand Canyon?

Updated as suggested

33-35: I realize this is a broad summary of previous work, but Cook et al. 2009 find rates of 300-500 m/myr, higher than the estimates in the previous sentence.

Sentence added to invoke Cook et al. 2009's rates specifically.

Figure 3d: Can you specify which terrace levels these are, in relation to figure 4? Are these Q2 and Q1?

Added as requested.

171:  Formatting of Nishiizumi et al (2007).

Fixed

190: I realize later you discuss the assumption of not terrace erosion; because fig3d sure makes me think there has been terrace erosion, I suggest you put a mention here that you'll address this assumption later.

Thanks, we have updated with a clause "…no terrace erosion (addressed in Section 4.3)…"

207:  I realize you address the point of sand deposits on top of the gravel later as well, but I got stuck on it here. Suggest you also add a sentence or statement that you discuss the specific stratigraphic relations between underlying gravel and overlying sand deposits, and how that influences interpretations of dates and rates, below.

We agree this is important to highlight as there are complicated stratigraphic relationships that result from using depositional (IRSL) and exposure age (cosmogenic isotopes) techniques on different units. Our attempt to clarify that we will address this is as follows: "In combination with the cosmogenic isotope depth profile, which dates the exposure of the gravel layer after abandonment, we can better constrain stratigraphic relationships and timings of terrace fill deposition and channel abandonment (see Section 4.3)."

250:  What is the "this may be the only location" statement based on?  Seems pretty interpretive; try to work this point into the discussion instead of here.

Adjusted scope as follows to limit speculation: "This is the only documented location along the San Juan River where such a local flow reversal has occurred."

266:  Clarify what the 14+-4 rate is based on, since its lower than the long-term incision rate. Is this based on the inheritance, representing the upstream basin at that time? Not everyone is as well-versed in the depth-profile methods as the authors; I suggest guiding readers like me a little more through these numbers.

This is a good point, it can be confusing that there is both a terrace erosion rate (the rate of lowering of the exposed terrace surface of which we are calculating an exposure age) and an inferred incision rate that is calculated based on the terrace exposure age and its height above the modern river. We have attempted to clarify this by changing the phrasing in Section 3.2 where the variable is introduced. Additionally, we have rephrased this section to: "Our best-fit terrace surface erosion rate (not to be confused with the river incision rate) overlaps within error with the 14 ± 4 m Myr-1 best-fit rate from Wolkowinsky and Granger (2004), but is slightly lower than previously estimated."

282:  "while of the older" has an extra word or needs more editing.

Fixed

285:  Suggest not saying the measurements are "correctly ordered".  The underlying gravels dating to be younger than the overlying sand deposits isn't correctly ordered.  I like the interpretation of why, it just feels like saying they are correctly ordered is overselling the data.

This is a good point and could be confusing to future readers. We have updated that section as follows:

"The ages of sand deposition, as measured by p-IR IRSL, and gravel exposure, as measured by $^{10}$Be depth profile, are coherent, with deposition preceding exposure. This finding is somewhat counter intuitive, however, given the sands overlay the gravels. As the top of the sand…
…Nonetheless, the meander cutoff could have occurred no earlier than the age of deposition of the units that were bypassed (~40 ka), and no later than the exposure age of the gravel package (~28 ka), thus bracketing the age of the meander cutoff."

Figure 7 caption: reword to not have "showing" twice.

Updated per this and Reviewer #1's comment.

Discussion, 301:  remove "being"

Fixed

303:  In this landscape, in my experience steps are usually associated with strong and weak lithologic changes. The authors bring this up in the next paragraph, but are you sure the strath terraces don't also reflect that?

We believe the arrangement of down-stepping terraces on the inside edge of a meander bend indicates the role of meander migration; if resistant rocks dominated at a meander, the same sequence would be seen throughout more of the meander's length and plausibly in nearby straight portions of the river as well. Additionally, the rock units we are dealing with in the area downstream of Raplee Ridge have regular bedding but no significant rock strength differences.

There are large contrasts between rock strength in the Rico Formation which influence erosion in small gullies nearby (see Steelquist et al., 2023). However, we do not see significant strath preservation either up- or downstream where the San Juan River traverses the Rico Formation.

348:  Maybe change "sinuosity of around 7.6" to "local sinuosity of around 7.6"?

Updated as suggested.

355-365, or 415-420, or somewhere:  I appreciate the broader discussion of why erosion rates are variable, and I think the cutoff bedrock meander is a nice example of that.  There are many other papers the authors could cite that I think are relevant to sediment supply and channel

morphology influencing long-term calculated erosion rates; two that come to mind are Ouimet et al. (2009) and Gallen et al. (2015).  It felt to me like the discussion could be a little broader.

These are both useful examples of other studies that have documented incision rate biases and transience in river incision studies, we thank the reviewer for suggesting them. The section has been updated as follows:

"Meander cutoffs provide a novel mechanism by which this "Sadler effect" may occur, complimenting previous research on biases in incision rates from strath terraces (Gallen et al., 2015) and on stochastic events that alter channel geometry (Ouimet et al., 2008)."

Citations

Ouimet WB, KX Whipple, BT Crosby, JP Johnson and TF Schildgen (2008), Epigenetic Gorges in Fluvial Landscapes. Earth Surface Processes and Landforms 33, 1993-2009, doi: 10.1002/esp.1650.

Sean F. Gallen, Frank J. Pazzaglia, Karl W. Wegmann, Joel L. Pederson, Thomas W. Gardner; The dynamic reference frame of rivers and apparent transience in incision rates. Geology 2015;; 43 (7): 623–626. doi: https://doi.org/10.1130/G36692.1

---

## Author Response (AR2)

Response to editor:

We thank the editor, Jens Turowski, for suggesting a body of literature that was previously missing from our manuscript. We believe the revised manuscript does better justice to the existing research on bedrock river meanders in both the highlighted sections.

Of particular interest to us was Moore (1926), which specifically detailed many of the stretches of the San Juan River which we studied. As mentioned by Turowski, Moore posited higher lateral migration with increasing bed cover as well as the relative strength of the lithology forming the bank, noting that once rivers encountered more erodible rock the meanders were "effectively obliterated". This corresponds very well with our observations of in the area surrounding our sampling sites and are glad to be able to give proper credit to Moore's work.

We have amended Line 90 and Section 5.3 to include the majority of the suggested references and give better context to the history of thinking on lateral migration in bedrock channels. Thank you for the suggestions!